# Influence of Fiber Distribution and Orientation in the Fracture Behavior of Polyolefin Fiber-Reinforced Concrete

**DOI:** 10.3390/ma12020220

**Published:** 2019-01-10

**Authors:** Alejandro Enfedaque, Marcos G. Alberti, Jaime C. Gálvez

**Affiliations:** Departamento de Ingeniería Civil: Construcción, E.T.S de Ingenieros de Caminos, Canales y Puertos, Universidad Politécnica de Madrid, C/Profesor Aranguren, s/n, 28040 Madrid, Spain; alejandro.enfedaque@upm.es (A.E.); marcos.garcia@upm.es (M.G.A.)

**Keywords:** fiber-reinforced concrete, polyolefin fibers, fiber distribution, fracture behavior, structural fibers

## Abstract

Polyolefin fiber-reinforced concrete (PFRC) has become an attractive alternative to steel for the reinforcement of concrete elements, mainly due to its chemical stability and the residual strengths that can be reached with lower weights. The use of polyolefin fibers can meet the requirements of standards, although the main constitutive relations are based on experience with steel fibers. Therefore, the structural contributions of the fibers should be assessed by inverse analysis. In this study, the fiber dosage was fixed at 6 kg/m^3^, and both self-compacting concrete and conventional concrete were used to compare the influence of the positioning of the fibers. An idealized homogeneous distribution of the fibers with such fibers crossing from side to side of the specimen was added to self-compacting concrete. The experimental results of three-point bending tests on notched specimens were reproduced by using the cohesive crack approach. Hence, constitutive relations were found. The significance of this research relies on the verification of the formulations found to build constitutive relations. Moreover, with these results, it is possible to establish a higher threshold for the performance of PFRC and the difficulties of limiting the first unloading branch typical of fracture tests of PFRC.

## 1. Introduction

Reinforced concrete was the most relevant construction material employed both in architecture and civil engineering during the 20th century. The widespread use of such material was widened in the second half of the century when fibers began to be added to common concrete formulations. Since that moment, the applications of fiber-reinforced concrete (FRC) have been drastically enlarged due to the development of a significant variety of fibers produced by combining materials, sizes, and shapes [1,2,3,4]. The improvement of the concrete properties that the fibers induce has enabled a large amount of uses, such as cracking control, fire spalling prevention, and multifunctional concretes [5,6,7], that enable applications such as guiding vehicles or heating pavements. In some of these uses, the metallic nature of steel fibers, which are the most common, might be an issue due to their potential corrodible and magnetic nature. In order to address such situations, certain types of polymeric fibers, which can be considered to be structural ones, have recently been developed. Those fibers provide structural capacities with lower dosages and are essentially polyolefin-based macrofibers. Moreover, such polyolefin fibers (PF) reduce the overall cost of the material and are chemically stable. Relevant research has been developed regarding the characterization of the mechanical properties of polyolefin fiber-reinforced concrete (PFRC) [8,9,10,11,12,13,14,15,16,17] and its engineering applications [18,19,20,21,22].

Fibers are added in the final stages of FRC production. Pouring methods, formwork geometry, and rheological properties of concrete add variations in the positioning and orientation of the fibers, as examined in References [23,24,25]. Such parameters have been studied through employing a wide variety of means, for example, the use of stereological tools [26], statistical ones [27], and even rheological analysis [28,29]. Most of the studies that deal with this matter have counted the amount of fibers placed in a sawn surface or in the fracture surfaces generated in bending fracture tests [30,31,32]. Krenchel proposed an orientation factor that couples the orientation and the distribution of the fibers, which permits analysis in these cases [33]. With the use of the orientation factor *θ*, it has been possible to determine the improvements in mechanical properties of steel fiber reinforced concrete (SFRC) by using self-compacting concrete (SCC) [34] and the effects of several types of vibration on conventional concrete (VCC) [30]. Nevertheless, at the time of writing, there is a gap in experimental data and analytical models for predicting the upper threshold of the improvement that a certain amount of polyolefin fibers, if ideally oriented and distributed, can provide to a certain type of concrete.

The main improvement that fibers provide to concrete appears after concrete cracking has taken place. Consequently, such post-cracking behavior has become the reference property, with several tests and recommendations being established [35,36,37,38,39,40,41] in order to enable a trustworthy comparison between the possible combinations of concrete and fibers. If the results comply with certain requirements established in the aforementioned recommendations, the contribution of the fibers added to the composite material might be taken into account. Therefore, it would be possible to reduce the amount of traditional steel rebars in reinforced concrete. Although this procedure is currently used, the development of predictive models that may allow the prediction of the number and positioning of the fibers placed in the critical sections of a structural piece is welcomed [34,42,43,44,45,46]. Such models might be a valuable tool for structural designers: They would also entail merging laboratory test results and everyday structural calculations.

Following this rationale, it would be even more significant to reproduce the fracture behavior of FRC, and especially PFRC due to the novelty of the material, by means of numerical simulations. This study adopted the cohesive crack approach. Since the fracture behavior of PFRC shows hardening, trilinear curves were used. Such a type of curve has been used for SFRC in cases of intermediate branches with soft-unloading or even flat stages [47,48]. However, the need for a branch with recharging strength has recently entailed an accurate reproduction of the mechanical behavior of PFRC [49,50]. This contribution was able to predict the mechanical response of PFRC with a determined amount of fibers by changing the parameters that defined a trilinear softening function. The variations of such parameters were performed based on several assumptions and by performing an inverse analysis. By combining the parameters obtained in the iterative analysis with a numerical regression, the functions that defined the parameters of the softening functions were defined. However, such assumptions and the functions that were deduced have not been corroborated yet.

In order to address such shortcomings in the field of PFRC, an experimental study was performed with self-compacting concrete and conventional vibrated concrete with a fiber dosage of 6 kg/m^3^ of 60-mm-long polyolefin fibers randomly distributed and named SCC6-60 and VCC6-60, respectively. Additionally, SCC with the same amount of fibers ideally positioned with a homogeneous distribution in the cross-section of the concrete elements was manufactured and termed SCC6-430. Previous experience in assessing the fracture behavior of PFRC with different amounts of polyolefin fibers suggested that the sensitivity of the distribution of fibers is lower for dosages of fibers equal to or above 10 kg/m^3^. Moreover, for dosages of polyolefin fibers of 4.5 or 3 kg/m^3^, there were occasions when there were only a few fibers in the fracture surface. Such phenomena hampered the assessment of the influence of the fiber orientation and distribution of fibers in the fracture behavior of concrete. Accordingly, a medium dosage of polyolefin fibers (6 kg/m^3^) was employed in this research. The previous information might be of interest for both future research and everyday production. The reinforcement capacity of the fibers added was assessed by comparing the behavior of the material when subjected to three-point bending fracture tests for the three compositions manufactured. In addition, the influence of the orientation factor was determined by counting the fibers placed in the fracture surfaces generated. Moreover, the test results, including those obtained in the specimens with an ideal homogeneous distribution of fibers, were reproduced by using the constitutive fracture model shown in Reference [49]. The changes added to the parameters that defined the softening function were compared for the formulations and the assumptions performed, and the functions that defined such variations were analyzed. The significance of this research lies in the definition of the threshold of the mechanical reinforcement of PFRC when the fibers added were ideally distributed. Consequently, structural designers can be aware of the maximum improvement of the properties that can be obtained using a determined dosage of fibers in PFRC. This entails the optimal dosage of fibers to be employed being determined with a high degree of accuracy. In addition, the importance of the distribution of the fibers was found when comparing the post-peak behavior of SCC6-60 and the SCC6-60 with the behavior of SCC6-430. This offers an extra safety factor when deciding the characteristics of the concrete and the production conditions that might influence both the orientation and distribution of fibers. Moreover, the assumptions made in References [8,24], which established a relationship between the maximum post-peak load (*L_REM_*) and the number of fibers in the fracture surface generated, were validated. Subsequently, such relations can be used in everyday practice with a higher degree of reliability. Besides, the numerical simulations performed validated the hypothesis assumed in the literature and enable a confident use of the functions proposed. Thus, the accuracy of the numerical simulations and the inverse analysis can improve future modeling procedures and supply reliable constitutive relations for PFRC related to physical parameters.

Summarizing, this contribution intends to reduce some of the uncertainties that still exist regarding the influence of the amount of fibers used and their distribution and orientation with the mechanical properties and constitutive relations of PFRC. In solving this issue, a more confident and reliable use of PFRC might be achieved, thus widening the field of applications of the material.

## 2. Concrete Production

Both vibrated concrete (VC) and SCC were manufactured with siliceous aggregates composed of two types of gravel of 4–8 mm and 4–12 mm and sand of 0–2 mm. The maximum aggregate size was 12.7 mm. In this study, a Portland cement type EN 197-1 CEM I 52.5 R-SR 5 [51] was employed. Limestone powder was used as a microaggregate with a specific gravity and a Blaine surface of 2700 kg/m^3^ and 400–450 m^2^/kg, respectively. The calcium carbonate content of such powder was higher than 98%, with less than 0.05% being retained in the 45-µm sieve. In both the SCC and VC, a polycarboxylate-based superplasticizer with a solid content of 36% and a density of 1090 kg/m^3^, called Sika Viscocrete 5720, was employed. The changes in the superplasticizer contents helped to obtain self-compactability in the fresh state behavior of SCC6-60 and SCC6-430. The formulations of VCC and SCC can be seen in Table 1.

In this study, the polyolefin fibers employed boasted two different lengths: First, 60-mm-long commercial fibers, and second, the same fibers but 600 mm long. Both fibers had the same mechanical and chemical properties and only differed in their length. The outlook of the 60-mm-long fibers where their rough surfaces appeared can be seen in Figure 1. Likewise, the mechanical properties supplied by the manufacturer can be seen in such an illustration.

With the aforementioned materials, three types of concrete were prepared. The first one was a vibrated conventional concrete and the last two were self-compacting concrete. In one SCC and in VCC, 6 kg/m^3^ of the 60-mm-long polyolefin commercial fibers previously described was used. In addition to VCC6-60 and SCC6-60, other specimens were manufactured with the same volumetric fraction of fibers but choosing the position and orientation of the fibers. This batch was named SCC6-430. In SCC6-60, a standard mixing procedure was used. The fibers were added in three stages and consequently were mixed with the rest of the components, leading to a random positioning of the fibers within the bulk. Once the VCC6 concrete was prepared, the material was poured in 430 × 100 × 100 mm^3^ molds following the recommendation of References [37,39]. In the case of SCC6-60, the fresh material was poured from one of the sides of the mold to allow the fibers to align in the direction of the flow.

In the case of SCC6-430, the fibers were positioned by choosing a homogeneous distribution and a perpendicular angle between the cross-section and the fibers. The amount of fibers positioned in the specimens was selected by taking into account the volumetric fraction that entailed a 6-kg/m^3^ addition of fibers being homogeneously distributed in the ligament of the cross-section. Consequently, as the volumetric fraction that corresponded to such an addition was 0.66%, the amount of fibers in a 100 × 100 m^2^ section was 66. The disposition of the fibers in the middle section of the specimens can be seen in Figure 2. The homogeneous distribution of fibers in the ligament of the cross-section differed from those found in real practice, as was determined in References [18,19,20,21,22,23,24,37,38]. Such homogeneous distribution was considered numb to the pouring method, the fresh-state behavior of concrete, and the wall effect, and consequently the fibers were distributed homogeneously. Although it is true that a better fracture behavior could be obtained if all the fibers were placed in the two lower thirds of the ligament, it should not be forgotten that such a distribution would be highly rare and quite far from everyday practice. No fibers were positioned in the bottom third of the specimen because such a location coincided with the position and depth of the notch. The self-compacting concrete was mixed without adding any fiber and poured in the molds without moving or causing any damage to the long fibers’ position. The four-step process can be seen in Figure 3.

In addition to the prismatic specimens, nine cylindrical specimens were prepared with a 150-mm diameter and a 300-mm height for each batch. All the specimens were stored in a climatic chamber at 20 ± 2 °C and above 95% humidity until the time of testing.

## 3. Material Characterization

The formulations were tested in order to obtain their main mechanical properties, such as compressive strength, indirect tensile strength, and modulus of elasticity. Such tests were performed according to the following recommendations: EN 12390-3:2009 (compressive strength), EN 12390-6:2009 (indirect tensile strength), and EN 12390-13 (modulus of elasticity). Table 2 shows the mechanical properties of all the formulations performed. Three tests were performed for obtaining the compressive strength and the indirect tensile strength. The rest of the specimens were employed for assessing the modulus of elasticity.

The mechanical properties shown in Table 2 showed slight differences among the concrete types used. All the results of the tests of the same formulation showed hardly any scattering, and therefore such results were considered statistically valid. While the compressive strength of VCC6-60 was 16% lower than that of SCC6-430, in the case of SCC6-60, this property was 6% higher than that of SCC6-430. In the case of the modulus of elasticity, VCC6-60 and SCC6-60 displayed values 17% and 12% lower than that of SCC6-430. The indirect tensile strength of VCC6-60 was only 1.5% lower than that of SS6-430, while in the case of SCC6-60, it was 8% higher than that of SCC6-430. Negligible differences were detected in the case of the fracture energies of the three formulations tested.

RILEM TC-187 SOC [53] was taken as a reference to carry out fracture tests in three prismatic specimens for each formulation. The setup of the three-point bending test, according to the standard noted above, was made in accordance with the depth of the beam. A span-to-depth ratio of 3.0 and a notch in the center of the span of 1/3 of the depth were chosen. The loading cylinder was placed above the notch. The test geometry setup is shown in Figure 4.

A crack-mouth opening displacement control (CMOD) of the test was performed using a clip-on gauge device (Instron 2620-602, Norwood, MA, USA). Two linear variable differential transformer (LVDT) devices (Instron 2601-044, Norwood, MA, USA) measured the deflection of the sample during tests, being placed on each side of the specimen. During the test, time, load, deflection, CMOD, and also the machine actuator position were recorded. Such a setup and procedure was used for testing the specimens of all formulations. An image of the testing procedure applied to a SCC6-430 specimen can be seen in Figure 5. The notch was machined with a water-cooled low-speed diamond cutting disc. The specimen positioning was carefully made by means of laser devices. The concrete beams rested on two rigid steel cylinders lying on two ground supports, which allowed free rotation out of the plane of the beam and guaranteed negligible friction rolling in the longitudinal direction of the beam. Similarly to what was seen in previous works [8,49,50], the results showed a low degree of scatter.

## 4. Fracture Tests Results

Regardless of the fiber rupture, fiber bridging, fiber pull-out, fiber debonding, or matrix cracking, these mechanisms prompted much higher deformations in all the specimens tested than in concrete and, consequently, the upper bound of the CMOD device was exceeded. The test continued changing the control parameter to the machine actuator position until deflection values up to 15 mm were reached. All the tests were stopped without reaching the failure of the specimens.

The curves obtained from the tests, shown in Figure 6, could be divided into three distinct trends that could be easily identified. All could be defined by changes in the load-bearing capacity of the material. The first one began at the start of the test and ended at the load at the limit of proportionality. Once the deflection at this point was surpassed, an unloading branch could be perceived. The shape of the unloading branch of the curves resembled the analogue part of the softening curve of conventional concrete. The unloading took place until fibers were capable of absorbing the energy released in the fracture processes. The fibers added to the concrete were able to sustain higher loads and, consequently, the load-deflection curve showed a positive slope after the minimum post-peak load. This slope was constant from a certain deflection, and afterwards, as certain damage mechanisms appeared, it decreased steadily until the maximum post-cracking load was reached. After reaching the maximum post-peak load, the load-bearing capacity of the material decreased progressively due to the continuation of the damage in the material. Consequently, the load-deflection curve showed a stable reduction of the mechanical capacity of the material until the end of the test was reached.

The aforementioned characteristics are common for the three formulations manufactured. The mechanical response of SCC6-60 was slightly better than VCC6-60, though no major disparities were found. In addition, such differences have already been reported in previous studies [10] and, in the case of this contribution, will be analyzed by means of the orientation factor in the following sections. Nevertheless, if the fracture curves of SCC6-430 are compared to analogous VCC6-60 and SCC6-60, remarkable features should be outlined.

There were no changes in the peak load obtained in the fracture tests. This result is coherent because there were only slight changes in the concrete formulations used. In addition, such results show that the different pouring method employed in the SCC6-60 and SCC6-430 specimens did not have any influence. Another point that should be highlighted is that in all formulations, the minimum post-peak load was similar and at around 30% of the peak load.

The slope of the reloading part of the curves in the case of the SCC6-430 specimens was noticeably greater. Whereas in the case of the VCC6-60 and SCC6-60 specimens, the maximum post-peak load was around 50%, when the curves of SCC6-430 are analyzed, it can be seen that this point reached almost the value of the peak load.

In addition, the deflection at the maximum loading capacity changed from 3.5 mm to 6.0 mm. It has to be emphasized that this observation has already been perceived to a lesser extent in previously published literature [23,24]. When reaching this deflection, different damaging mechanisms appeared with dissimilar importance. In the case of the concrete formulations performed with short fibers, the following damage mechanisms appeared: Matrix cracking, fiber-matrix debonding, fiber pull-out, and fiber rupture. However, the changes in the slope that are perceived when comparing the SCC6-60 and VCC6-60 curves to the SCC6-430 curves could be attributed to the damage mechanisms that could not appear when reinforcing with long fibers: Fiber pull-out. Another point worth mentioning is the lack of scattering in the case of the SCC6-430 specimens when compared to the specimens performed with short fibers. This observation explains that the inherent scattering that appeared in the fracture tests of the FRC was caused by the differences in the amount of fibers and their positioning in the fracture surface.

Analyzing the final unloading part of the tests, it can be seen that the unloading process seemed to be more gradual in the case of the SCC6-430 specimens. However, a deeper study would be required to obtain sound conclusions. In any case, as in previous studies by the same authors in References [8,10], the characteristic points of the experimental mean curves are extracted in Table 3 in order to ease their discussion.

## 5. Fracture Surface Analysis

In order to quantify the importance of the orientation and distribution of the fibers, an analysis of the fracture surfaces generated in the three-point bending tests was required. In the case of the SCC6-430 specimens, the amount of fibers was predetermined in the manufacturing process to equal the theoretical one, with them having a perfect distribution and positioning. However, a fiber-counting exercise was required to assess the number of fibers in the fracture surfaces of the rest of the formulations. Figure 7 shows the appearance of two specimens of VCC6-60 and SCC6-60: The notch required to perform the fracture test can be clearly seen.

The theoretical number of fibers placed in the fracture surface was obtained for each concrete by using Equations (1) and (2), considering that the fibers were uniformly distributed and perpendicular to the crack:(1)$$V_{f}=\frac{W_{f}}{\rho .V},$$
(2)$$th=\frac{A\;V_{f}}{A_{f}}.$$

In Equations (1) and (2), *V_f_* is the fiber volumetric fraction, *W_f_* is the weight of the fibers for a reference volume of 1 m^3^, *ρ* is the fiber density, and *V* is the total volume. Moreover, *th* is the theoretical number of fibers that appear in the fracture surface of a given specimen, with *A* being the cross-section of the specimen and *A_f_* the section of one fiber. The average total number of fibers obtained can be seen in Figure 8. Furthermore, the relation between the fibers counted in a given cross-section (*n*) and the theoretical number of fibers (*th*) are also shown in Figure 8. This relation *θ*, which assumes a homogeneous distribution of fibers, has been called the orientation factor in previous research [33]. The previous relation can be observed in Equation (3):(3)$$\theta =\frac{n}{th}=n\frac{A_{f}}{V_{f}A}.$$

In the case of the SCC6-430 specimens, the theoretical fiber content was homogeneously distributed in all the specimen sections, and consequently, in such specimens the value of the orientation factor was 1. The evaluation of the orientation factor was performed using the approach aforementioned, and the results are also shown in Figure 8, including the percentage of pull-out fibers.

## 6. Numerical Simulations

Similarly to the process followed in References [49,50,54], the fracture results obtained in the experimental campaign were reproduced by means of numerical simulations. This process was carried out by using the commercial software Abaqus (Abaqus 6.13, Dassault Systems, Vélizy-Villacoublay, France) through merging the features implemented in such a code with a material user subroutine.

The fracture behavior was reproduced by employing 2D numerical models meshed through using three-node triangular elements with one Gauss point. The mechanical behavior of the material was linear elastic without any damage when under compression. The tensile behavior of the material was linear elastic until the tensile stress was reached. If the strain that corresponded to the tensile strength was surpassed, the mechanical response of the material would be governed by a softening function implemented in a user material subroutine. In previous studies, it was shown that a trilinear softening function was suitable for reproducing the fracture behavior of the material, with the same approach being used in the case of the specimens studied in this paper. The shape and the characteristic points that defined the softening function can be seen in Figure 9.

The nonlinear fracture process zone appeared in the elements placed on the crack. The behavior of the fracturing elements depended on a constitutive relation that needed to be iteratively fit until finding the values of *C_MIN_*, *C_REM_*, and *C_F_* that were capable of reproducing the fracture behavior of all three formulations with a reasonable degree of accuracy.

The inverse analysis used has been explained in depth in previous papers [49]. The mechanical data of the material that was obtained in the mechanical test and used in the simulations can be seen in Table 2. For the specific fracture energy (*G_F_*) and the Poisson coefficient of plain concrete, values of 130 N/m and 0.2 were respectively adopted [49].

With the aforementioned data and by exerting the inverse analysis previously cited, it was possible to find an accurate reproduction of the experimental tests. In addition, the trilinear softening functions were defined. In Figure 10, both numerical results and experimental results can be seen.

Although there were variations in the mechanical properties of the formulations tested, for the numerical simulations the values of the compressive strength, modulus of elasticity, and Poisson’s ration remained constant. Although it could be argued that such absence of variations might have resulted in certain inaccuracies, it should be considered that the main subject of these contributions dealt with the contribution of the fibers added and their influence on the fracture behavior. Moreover, when the influence of the fibers appeared in the mechanical behavior, there was scarcely any influence of the mechanical properties previously mentioned.

The curves that appear in Figure 10 clearly reproduce, and with a significant degree of accuracy, the experimental results. It should be highlighted that the accuracy of the simulations in the cases of SCC6-60 and VCC6-60 was notable before the post-peak maximum load was reached. However, this similarity was reduced from this point onwards. The differences that appear between the experimental curves and the numerical ones were caused by the unpredictable nature of the appearance of the aforementioned damage mechanisms. This was clearly confirmed when the scattering of the experimental curves was more reduced, as in the case of the SCC6-430 specimens. It can be clearly seen that the curves obtained by the numerical simulations in this case fit the experimental results extremely well.

The reproduction of the mechanical behavior of SCC6-60 and VCC6-60 implied changes in *f_ct_*, *C_MIN_*, and *C_REM_*. Regarding those of *C_MIN_* and *C_REM_*, it can be seen in Table 4 that there were only slight changes. In the case of *C_F_*, the value was stable for both formulations. In order to reproduce the tests of the SCC6-430 specimens, we needed to change the value of *C_MIN_* slightly. Nevertheless, in the case of *C_REM_*, not only the stress value had to be modified. The crack opening at which *C_REM_* took place changed noticeably in the specimens with the fibers homogeneously distributed and shifted from a value of 2.25 to 2.75. Similarly, the value of *C_F_* had to be fixed at 12.5 in order to obtain an accurate reproduction of the experimental results. The outlook of the softening functions implemented can be seen in Figure 11.

## 7. Discussion

This section deals with the relevant aspects of the experimental results, connections between the orientation and distribution of the fibers with such results, and changes that were performed in the material subroutine implementation in order to find an accurate reproduction of all the tests.

As could be seen in the corresponding section, the general shape of the fracture curves obtained for the formulations were analogous. Both SCC6-60 and VCC6-60 were highly similar to each other. Moreover, there were significant similarities among the fracture curves of SCC6-60, VCC6-60, and SCC6-430. Although this can be seen in Figure 6, there are several aspects that should be underlined. First, in all formulations, the minimum post-cracking loads were similar. This feature is of primary importance because it clearly states that the bearing capacity for small deformations did not depend on the positioning or orientation of the fibers, because the orientation factor of the VCC6-60 specimens on average was 0.72 and in the case of SCC6-60 was 0.78, which was noticeably smaller than the unity that the SCC6-430 specimens boasted. In addition, as can be seen in Figure 8 and Figure 12, although the coefficient of variation was 0.25 between the amounts of fibers in the lower third of the fracture surfaces generated, there were no remarkable changes of the value of *L_MIN_* recorded. In addition, this tendency fit the previous data available in literature. Therefore, as stated in Reference [49], such a load value is mainly influenced by the amount of fibers added, and the rest of the parameters might be considered to have only a limited impact.

Another point that should be highlighted is that the slope of the reloading part of the curves is noticeably greater in the case of the SCC6-430 specimens. The slope of the SCC6-430 formulation in this part of the experimental curve is 181% higher than in the case of the VCC6-60 formulation and 156% greater than in the case of the SCC6-60. However, there was no clear relation between such increments and the difference in the presence of fibers in any of the combinations obtained by adding the amount of fibers in the thirds of the ligament surface.

Regarding the maximum post peak value, *L_REM_*, it should be underlined that the experimental values obtained in the curves of the VCC6-60 and SCC6-60 specimens were above the values obtained in the literature. The linear tendency that related the amount of fibers and *L_REM_* seemed to be valid. Conversely, the value obtained in the case of the specimens manufactured with the 430-mm-long fibers was clearly above the values or even the predictions that might have been applicable to all the formulations performed with short fibers of the same characteristics. It could be argued that there were a greater number of fibers in the ligament of the specimen and thus the load that the specimens were able to sustain was also greater. Following this rationale, in the whole section there was an amount of fibers 128% and 111% greater in the case of SCC6-430 when compared to VCC6-60 and SCC6-60, respectively. However, if the values of *L_REM_* obtained for SCC6-430 were multiplied by the coefficient of orientation, the *L_REM_* values of VCC6-60 and SCC6-60 should have been 4472 N and 4071 N. Such values are considerably higher than those obtained. Consequently, the *L_REM_* values achieved by the formulations performed with 60-mm-long fibers should have shown damage mechanisms that only appeared when short fibers were added. One of these mechanisms could have been the pull-out of the fibers (see Reference [55]), which played a major role in the composite material behavior. Therefore, it is important to highlight that not only the orientation factor reduced the load-bearing capacity of the specimens in VCC6-60 and SCC6-60 formulations.

In the case of the unloading branch of the curves, it can be seen that the unloading process was steadier in the case of the SCC6-430 specimens. Whereas in the cases of SCC6-60 and VCC6-60 the decrement of the load-bearing capacity was noticeable from *L_REM_* onwards, in the case of the VCC6-430 specimens it seemed that this process was slower. It should be remembered that in the VCC6-430 specimens there was no possible pull-out of the fibers, whereas this damage mechanism appeared in a certain amount of fibers present in the fracture surface of SCC6-60 and VCC6-60 specimens.

Regarding the numerical analysis performed, it can be concluded that only by applying minor changes in the trilinear softening function found in the literature [49] was it possible to reproduce both the experimental results of the VCC6-60 and SCC6-60 specimens. However, several changes had to be applied in order to simulate the behavior of the SCC6-430 specimens. Whereas in the cases of the VCC6-60 and SCC6-60 specimens the changes of the values of *C_REM_* were limited to the strain value, in the case of the SCC6-430 specimens it was necessary to also change the crack width where the *C_REM_* took place. Moreover, the value at which *C_F_* took place had to be modified in a noteworthy way. This was in accordance with what has been proposed in previous studies [11,49,50].

Concerning the functions derived in References [11,49], the inverse analysis performed in this contribution enabled checking of the validity of the equations deduced. If Equation (4) is considered, it can be seen that the only parameter of the material was the volumetric fraction and, consequently, all the predictions of the *C_MIN_* value were unique for all formulations. In Table 5, the values obtained by means of the combination of the inverse analysis and the numerical simulations are shown. As can be seen in this table, the predicted value through using Equation (1) provided an accurate fit of *C_MIN_* that corresponded to VCC6-60:(4)$$\Phi =-3.6046+5.0625\left(1-e^{6.55Vf}\right).$$

However, this was not the case for SCC6-60 or even SCC6-430. If the *C_MIN_* value was considered to be correct, the volumetric fraction could be deduced. In the case of the two formulations, such a deduced volumetric fraction corresponded to a PFRC formulation with a 10 kg/m^3^ addition of fibers. This value could be explained if the homogeneous and artificially prepared disposition of the fibers was taken into account. Nevertheless, such an argument cannot be applied to the SCC6-60 result, where an equal volumetric fraction of 1% was obtained. Such a remarkable variation was not only caused by the number of fibers in the lower third of the ligament, but also by the positioning of the fibers and the improvements obtained in the distribution of the fibers due to the flux of self-compacting concrete [23]. In this lower third, the average position of the fibers was not in the center of gravity of such an area but closer to the tip of the notch. Consequently, the contribution of the fibers toward regaining load-bearing capacity was greater than in a normal situation because on average their strain was greater than was the strain borne by them.

Considering the value of stress at *C_REM_*, which could be obtained by means of Equation (5), two conditions should be pointed out. Whereas in the case of VCC6-60 and SCC6-60 the amount of pulled-out fibers and their distribution had to be found, with both formulations being close to 20% and 0.72 and 0.79, respectively, in the case of the SCC6-430 specimens the value of fibers pulled out was zero, while the value of *θ* was 1. By applying these conditions to Equation (2), the value of *C_REM_* for all the formulations can be seen in Table 5. In the case of SCC6-60, there was an accurate prediction of the values used. However, in the case of the other two formulations, there was a deviation of up to 18%. Equation (5) is
(5)$$\sigma_{CREM}=\left(1-\%\;pulled-out\;\right)Vf\;\theta \;\sigma_{u}.$$

Another point that should be highlighted is that with longer fibers, the deflection where *C_REM_* occurred changed. Such an idea was suggested when comparing the fracture test results shown in Reference [23], but due to the reduced difference in the fiber length, it was difficult to perceive. Nevertheless, when using 430-mm-long fibers, it was easy to confirm this phenomenon, even when there was a reflection in the softening function implemented. There was a shift of 0.5 mm in the case of the crack opening of *C_REM_*, and in the case of *C_F_*, there was a 5-mm shift. Consequently, it can be stated that the longer the fiber, the greater the value of *C_F_* that should be implemented.

## 8. Conclusions

The matter of the mechanical threshold of the reinforcement provided by a certain amount of polyolefin fibers remained unsolved due to the influence of two coupled parameters of PFRC: The fiber distribution and the fiber orientation. This study showed that if fibers were not pulled out, and they were homogeneously distributed, the mechanical capacity of the fibers added enabled reaching stress values close to those obtained in the limit of proportionality. It is also worth underlining that with such changes, the performance of only 6 kg/m^3^ of fibers was even superior to a formulation with a 10-kg/m^3^ addition of short fibers. The results highlight the importance of an even distribution of fibers within the fracture surface.

The analysis of the fracture tests performed confirmed that *L_MIN_* was essentially related to the amount of fibers present in the lower third of the ligament. Moreover, if *L_REM_* was considered, the behavior of the material was greatly influenced by the fiber length due to the damage mechanisms that appeared. It was shown that the amount of energy absorbed by the material greatly increased when long fibers were used. It has to be underlined that before this research was carried out, there were still uncertainties about the relations between the amount of fibers, their characteristics, and their distribution in the fracture surface and the distinctive points that define the fracture behavior of PFRC.

Regarding the numerical simulations, the importance of the distribution and orientation of fibers was clearly stated, as well as the need for models and formulations that help structural designers to consider these types of predictive tools and numerical results. The fracture tests were reproduced and provided a remarkable degree of accuracy. The results of both the numerical simulations and experimental results provide valuable discussion and verification of the parameters and expressions proposed in previous research [11,42] with regard to the influence of the coefficient of orientation.

## Figures and Tables

**Figure 1 materials-12-00220-f001:**
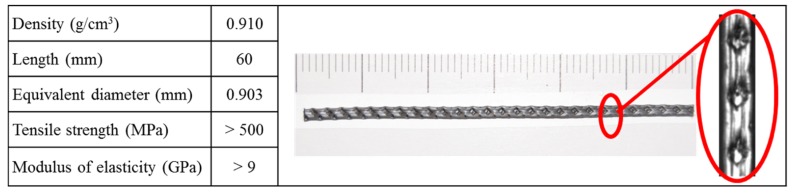
Commercial polyolefin fiber. Scale in mm. Mechanical properties supplied by the manufacturer (Sikafiber M60, Madrid, Spain).

**Figure 2 materials-12-00220-f002:**
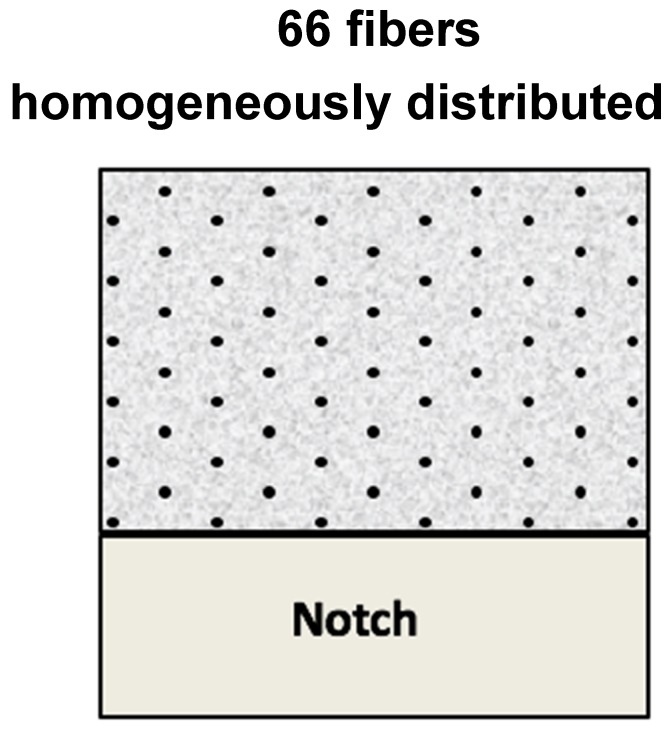
Cross-section of the specimens performed with a homogeneous distribution of fibers.

**Figure 3 materials-12-00220-f003:**
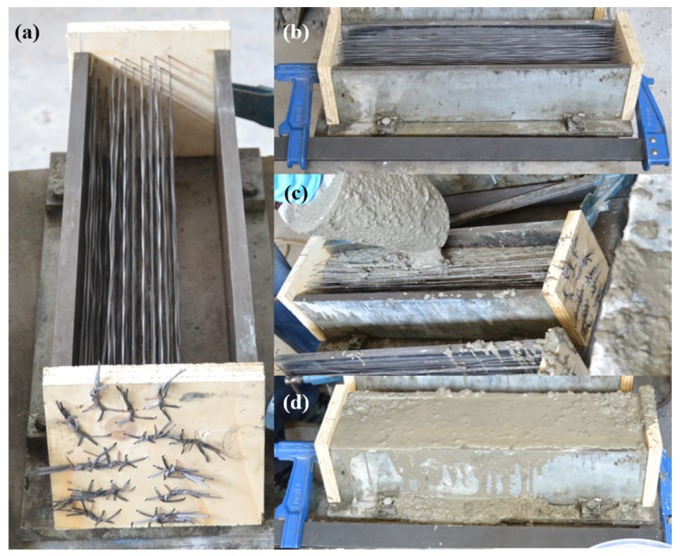
(**a**) Positioning of the long fibers in the molds; (**b**) fixing the mold sides to avoid any damage in the fibers while concrete was in a fresh state; (**c**) the pouring of self-compacting concrete; (**d**) final appearance of the specimens.

**Figure 4 materials-12-00220-f004:**
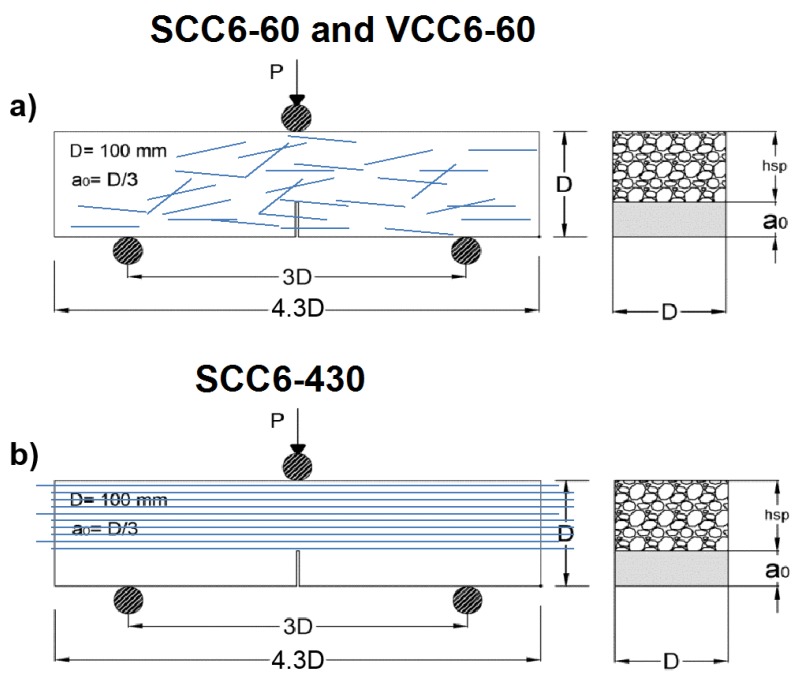
(**a**) Test geometry in the case of fibers randomly positioned; (**b**) test geometry in the case of fibers being aligned in the stress direction and homogeneously distributed.

**Figure 5 materials-12-00220-f005:**
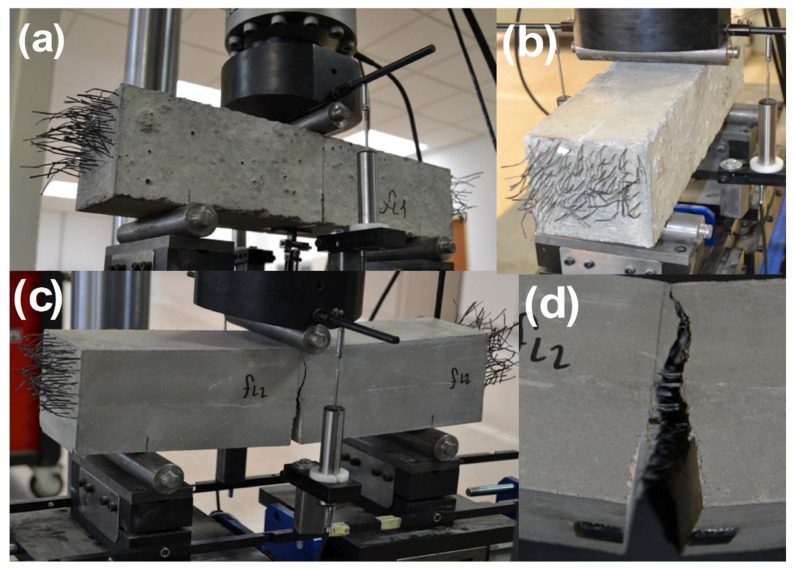
(**a**) Specimen ready for the test; (**b**) test disposition in the testing machine; (**c**) crack fully developed while testing; (**d**) image of the fracture surface generated in the fracture test with the fibers bridging the crack sides.

**Figure 6 materials-12-00220-f006:**
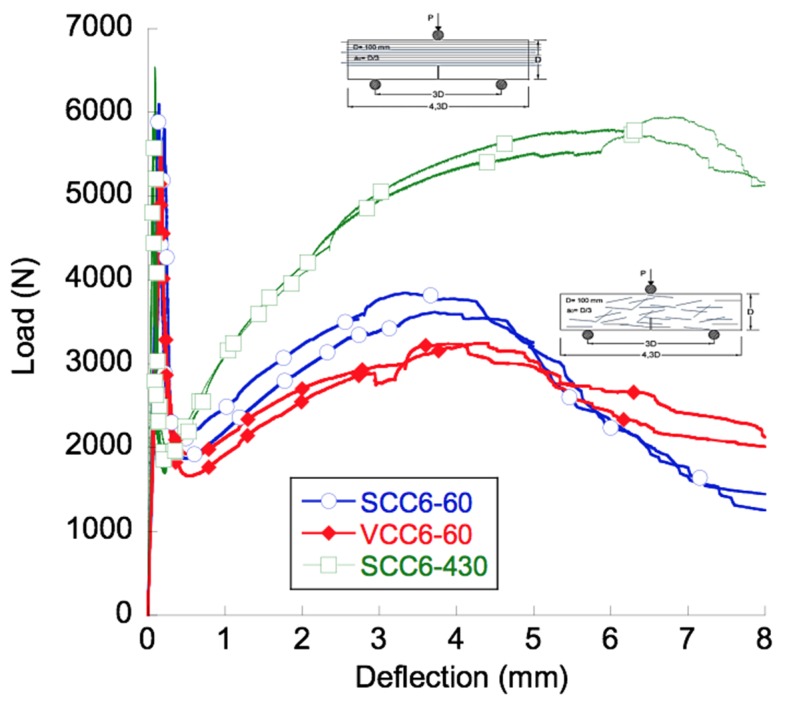
Three-point bending fracture test results.

**Figure 7 materials-12-00220-f007:**
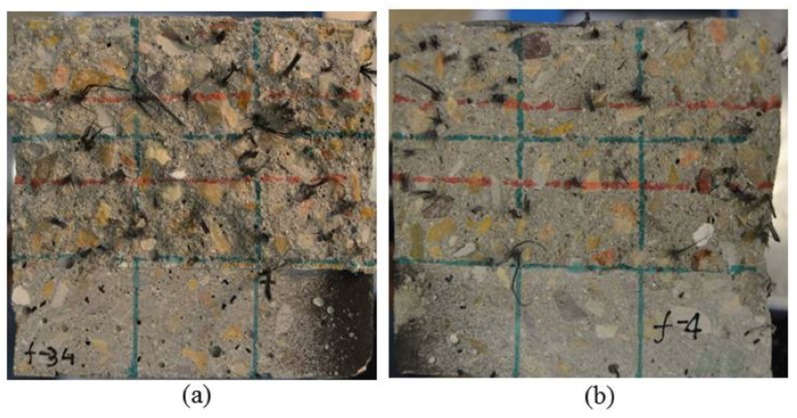
Fracture surfaces generated: (**a**) VCC6-60; (**b**) SCC6-60.

**Figure 8 materials-12-00220-f008:**
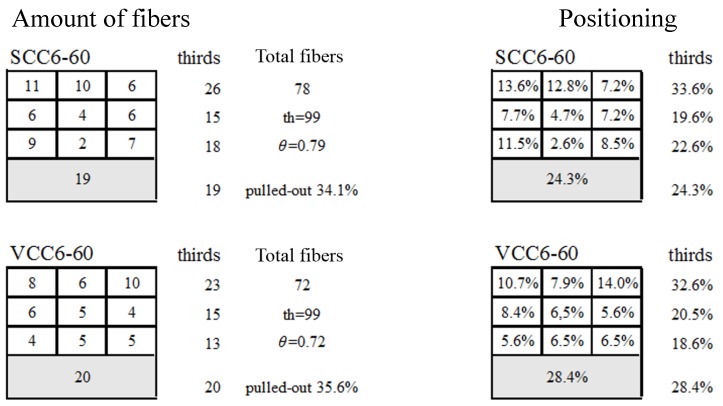
Counting exercise performed in SCC6-60 and VCC6-60 (including the percentage of pull-out fibers).

**Figure 9 materials-12-00220-f009:**
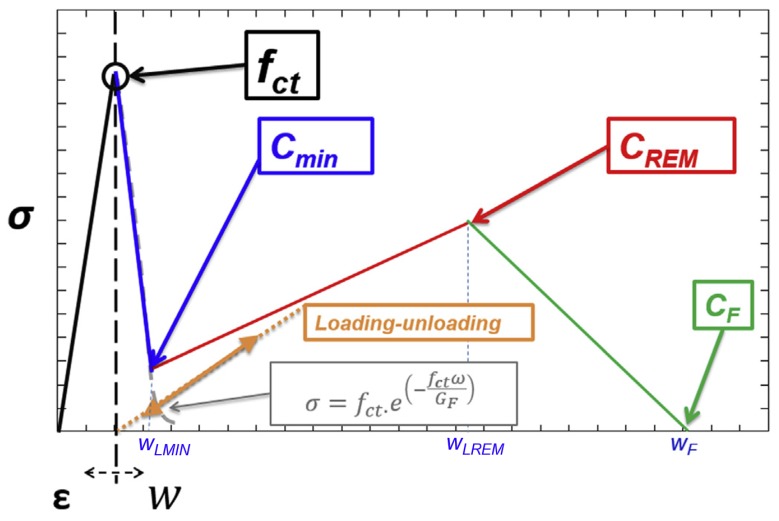
Sketch of the shape and position of the turning points of the constitutive relations for polyolefin fiber-reinforced concrete (PFRC).

**Figure 10 materials-12-00220-f010:**
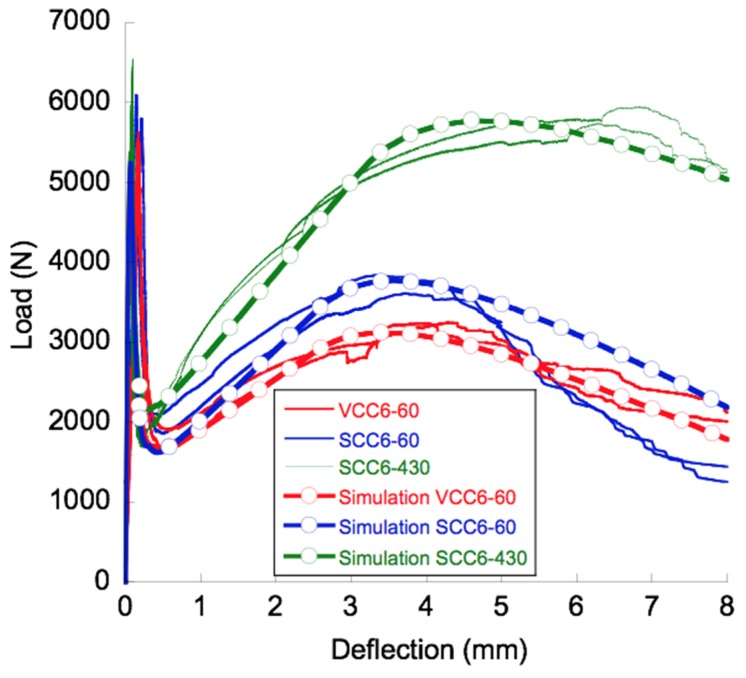
Results of the numerical simulations and the experimental tests.

**Figure 11 materials-12-00220-f011:**
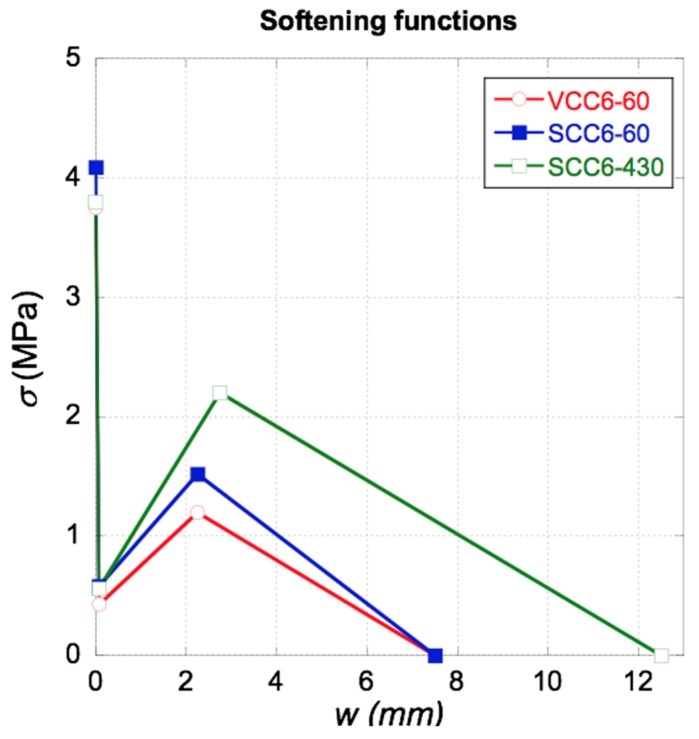
Softening functions implemented.

**Figure 12 materials-12-00220-f012:**
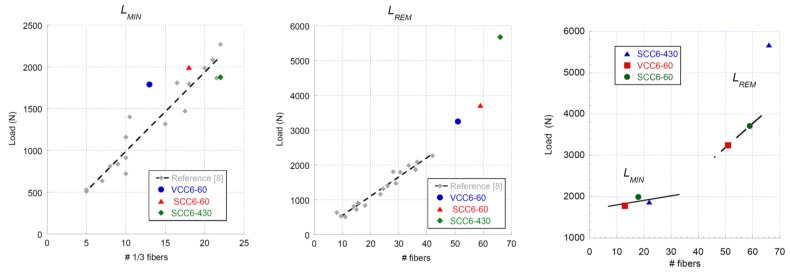
Relation between *L_MIN_* and *L_REM_* and the number of fibers in the ligament.

**Table 1 materials-12-00220-t001:** Concrete formulation per m^3^. SCC: Self-compacting concrete; VCC: Vibrated conventional concrete.

Concrete Type	Cement	Limestone Powder	Water	Sand	Gravel	Grit	Superplasticizer (% Cement Weight)
SCC6-60 and SCC6-430	375	200	187.5	918	245	367	1.25
VCC	375	100	187.5	916	300	450	0.75

**Table 2 materials-12-00220-t002:** Mechanical properties of the concrete formulations.

Mechanical Properties	VCC6-60	SCC6-60	SCC6-430
Compressive strength, *f_ck_* (MPa)	32.9	41.4	39.0
Modulus of elasticity, *E* (GPa)	29.9	31.6	36.0
Tensile strength, *f_ct_* (MPa)	3.75	4.09	3.80
Fracture energy *, *G_F_* (N/m)	131	130	130

* Values related to their corresponding plain concretes in Reference [52].

**Table 3 materials-12-00220-t003:** Minimum (*L_MIN_*) and maximum (*L_REM_*) post-cracking loads and their corresponding crack openings (w*_LMIM_* for *L_MIN),_* and w*_LREM_* for *L_REM_*).

Concrete Type	*L_MIN_* (kN)	w*_LMIM_* (mm)	*L_REM_* (kN)	w*_LREM_* (mm)
SCC6-430	1.875	0.22	5.676	6.1
VCC6-60	1.788	0.54	3.246	4.1
SCC6-60	1.994	0.45	3.712	3.6

**Table 4 materials-12-00220-t004:** Turning points for the numerical simulation.

Concrete Type	*C_MIN_*		*C_REM_*		*C_F_*	
	W (mm)	σ (MPa)	W (mm)	σ (MPa)	W (mm)	σ (MPa)
VCC6-60	0.08	0.43	2.25	1.20	7.5	0
SCC6-60	0.07	0.58	2.25	1.52	7.5	0
SCC6-430	0.07	0.56	2.75	2.20	12.5	0

**Table 5 materials-12-00220-t005:** Comparison of the predicted values in References [11,42] to those obtained in this study by inverse analysis.

Concrete Type	*C_MIN_*			*C_REM_*		
	Inverse Analysis	Predicted by Reference [49]	Δ (Variation)	Inverse Analysis	Predicted by Reference [42]	Δ (Variation)
VCC6-60	0.43	0.43	0.00%	1.20	1.42	−18.33%
SCC6-60	0.58	0.43	25.85%	1.52	1.57	−3.29%
SCC6-430	0.56	0.43	23.21%	2.20	2.48	−12.73%

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
