# Peer review of "Influence of Fiber Distribution and Orientation in the Fracture Behavior of Polyolefin Fiber-Reinforced Concrete"

_materials, 2019, doi:10.3390/ma12020220_

Reviewer 1 Report

The article entitled "Influence of the fibre distribution and orientation in the fracture behaviour of polyolefin fibre reinforced concrete" contains an interesting analysis in the field of materials science on the basis of reinforcing way of the cement concrete.  In this contribution was added a computer simulation using FEM some interesting expertise quantity analysis.  In reviewer opinion the article was written clear using corrected English language. Here are some minor comments:

1. Table 2 presents a comparative analysis of the mechanical properties of concrete samples. The authors used the term "slight differences" or "minor importance" and so on. I believe that to do this you should use some statistical tools. It should be remembered that the quantitative assessment can be made on the basis of the evaluation of the spread value, i.e. the difference of the result between the two samples. The authors should also more strongly indicate how many speciments were made in the experiment to evaluate these properties. In the current state, it is difficult to say whether the differences are or are not between averages. Of course, assuming that the estimator of averages was not biased by the asymmetry of the theoretical distribution of results.

2. Did the authors consider the optimal length of fibers to granulation of mineral mix?

3. I believe that the increase in Compressive strength and Tensile strength in case of SCC-60 was related to the change in Poisson's ratio (and thus the bulk elastic modulus). Did the authors consider this aspect in FEM modeling.

Author Response

The article entitled "Influence of the fibre distribution and orientation in the fracture behaviour of polyolefin fibre reinforced concrete" contains an interesting analysis in the field of materials science on the basis of reinforcing way of the cement concrete.  In this contribution was added a computer simulation using FEM some interesting expertise quantity analysis.  In reviewer opinion the article was written clear using corrected English language. Here are some minor comments:

1. Table 2 presents a comparative analysis of the mechanical properties of concrete samples. The authors used the term "slight differences" or "minor importance" and so on. I believe that to do this you should use some statistical tools. It should be remembered that the quantitative assessment can be made on the basis of the evaluation of the spread value, i.e. the difference of the result between the two samples. The authors should also more strongly indicate how many speciments were made in the experiment to evaluate these properties. In the current state, it is difficult to say whether the differences are or are not between averages. Of course, assuming that the estimator of averages was not biased by the asymmetry of the theoretical distribution of results.

The authors thank the reviewer for the comment as they think that a deeper explanation will improve the quality of the contribution. In accordance, the following wording has been added to the manuscript. 

The formulations were tested in order to obtain their main mechanical properties, such as compressive strength, indirect tensile strength and modulus of elasticity. Such tests were performed according to the following recommendations: EN 12390-3:2009 (compressive strength), EN 12390-6:2009 (indirect tensile strength) and EN 12390-13 (modulus of elasticity). Table 2 shows the mechanical properties of all the formulations performed. Three tests were performed for obtaining the compressive strength and the indirect tensile strength. The rest of specimens were  employed for assessing the modulus of elasticity.

The mechanical properties shown in Table 2 show slight differences among the concrete types used. All the results of the tests of the same formulation showed hardly any scattering and therefore such results were considered statistically valid. While the compressive strength of VCC6-60 was 16% lower than the one of SCC6-430 in the case of SCC6-60 such property was 6% higher than one of SCC6-430. In the case of the modulus of elasticity VCC6-60 and SCC6-60 displayed values 17% and 12% lower than the one of SCC6-430. The indirect tensile strength of VCC6-60 was only 1.5% lower than the one of SS6-430 while in the case of SCC6-60 it was 8% higher than the one of SCC6-430. Negligible differences were detected in the case of the fracture energies of the three formulations tested.

2. Did the authors consider the optimal length of fibers to granulation of mineral mix?

The authors would like to explain that such possibility was considered when planning the experimental campaign. However, it was decided to maintain the same aggregate skeleton in order to provide a meaningful comparison among the formulations tested.

3. I believe that the increase in Compressive strength and Tensile strength in case of SCC-60 was related to the change in Poisson's ratio (and thus the bulk elastic modulus). Did the authors consider this aspect in FEM modeling. 

The authors appreciate the comment performed by the reviewer as it points out a characteristic that deserves being explained in the manuscript. Consequently the following changes have been performed in the manuscript: 

Although there were variations in the mechanical properties of the formulations tested for the numerical simulations the values of the compressive strength, modulus of elasticity and Poisson’s ration were maintained constant. Although it could be argued that such absence of variations might result in certain inaccuracies, it should be considered that the main subject of these contribution deals with the contribution of the fibres added and their influence in the fracture behaviour. Moreover, when the influence of the fibres appeared in the mechanical behaviour there was scarcely any influence of the mechanical properties previously mentioned. 

 Reviewer 2 Report

The authors should better highlight in the abstract and introduction the RESEARCH SIGNIFICANCE of the manuscript.

Plastic fiber it's becoiming a prolific topic in construcction & engieenring journal. In order to guide readers, please include some further references of key authors on the topic of macro-plastic fiber reinforced concrete (real scale test - experiences on slabs - orientation - numerical studies - characterization - ... )

Please provide some research paths for other researchers and give some recommendations.

Rewrite conclusions focusing on the originality and core objectives of this work

This is a good paper!

Author Response

The authors should better highlight in the abstract and introduction the RESEARCH SIGNIFICANCE of the manuscript.

The authors greatly appreciate the kind comments of the reviewer. Moreover, his/her suggestions have been adopted and certain changes have been introduced in the manuscript. 

The final paragraph of the introduction has been modified trying to underline the importance of the research performed and the implications that the results might have in the present state of the art and in the everyday practice alike. The modified wording can be seen in the following lines:

The significance of this research lies in the definition of the threshold of the mechanical reinforcement of PFRC when the fibres addedare ideally distributed. Consequently, structural designers can be aware of the maximum improvement of the properties of that can be obtained using a determined dosage of fibres in PFRC. This entails that the optimum dosage of fibres to be employed can be determined with a higher degree of accuracy. In addition, the importance of the distribution of the fibres would be found when comparing the post-peak behaviour of SCC6-60 and the SCC6-60 with the behaviour of SCC6-430. This offers an extra safety factor when deciding the characteristics of the concrete and the production conditions that might influence the orientation and distribution of fibres alike. Moreover, the assumptions made in references [8, 17]that established a relation between the maximum post-peak load (LREM) and the number of fibres in the fracture surface generated are validated. In addition, the relation between the minimum post-cracking load and the amount of fibres in the lower third of the fracture surface generate is also validated. Subsequently, such relations can be used in everyday practice with a higher degree of reliability. Besides, the numerical simulations performed would validate the hypothesis assumed in the literature and enable a confident use of the functions proposed. Thus, the accuracy of the numerical simulations and the inverse analysis can improve future modelling procedures and supply reliable constitutive relations for PFRC related with physical parameters.

Summarising, this contribution intends to reduce some of the uncertainties that still exist regarding the influence of the amount of fibres used and their distribution and orientation with the mechanical properties and the constitutive relations of PFRC. Solving this issues a more confident and reliable use of PFRC might be achieved thus widening the field of applications of the material.

Plastic fiber it's becoiming a prolific topic in construcction & engieenring journal. In order to guide readers, please include some further references of key authors on the topic of macro-plastic fiber reinforced concrete (real scale test - experiences on slabs - orientation - numerical studies - characterization - ... )

The suggestion of the reviewer has been considered positive for the quality of the paper and consequently several references have been included in the manuscript. 

Please provide some research paths for other researchers and give some recommendations.

The authors reckon that the suggestion of the reviewer might be useful for readers and consequently have introduce several new sentences 

 Previous experience in assessing the fracture behaviour of PFRC with different amounts of polyolefin fibres have suggested that the sensitiveness of the distribution of fibres is lower for dosages of fibres equal or above 10kg/m³. Moreover, for dosages of polyolefin fibres of 4.5 or 3 kg/m³ there were occasions when there were only few fibres in the fracture surface. Such phenomenon hampers assessing the influence of the fibre orientation and distribution of fibres in the fracture behaviour of concrete. Accordingly, a medium dosage of polyolefin fibres (6kg/m³) was employed in this research. The previous information might be of interests for future research and everyday production alike

Rewrite conclusions focusing on the originality and core objectives of this work. 

The authors thank the reviewer for the comment and subsequently the conclusions have been summarised trying to highlight the most relevant findings of the research. The new wording can be seen in the following lines: 

The matter of the mechanical threshold of the reinforcement provided by a certain amount of polyolefin fibres remained unsolved due to the influence of two coupled parameters of PFRC: the fibre distribution and the fibre orientation. This study showed that if fibres are not pulled out, and they are homogeneously distributed, the mechanical capacity of the fibres added enable reaching stress values close to those obtained in the limit of proportionality. It is also worth underlining that with such changes the performance of only 6kg/m³ of fibres was even superior to a formulation with a 10kg/m³ addition of short fibres. The results highlight the importance of an even distribution of fibres within the fracture surface. 

The analysis of the fracture tests performed confirmed that LMIN was essentially related with the amount of fibres present in the lower third of the ligament. Moreover, if LREM is considered, the behaviour of the material was greatly influenced by the fibre length due to the damage mechanisms that appear. It was shown that the amount of energy absorbed by the material greatly increases when long fibres were used. It has to be underlined that before this research was carried out there were still uncertainties about the relations between the amount of fibres, their characteristics and their distribution in the fracture surface and the distinctive points that define the fracture behaviour of PFRC. 

Regarding the numerical simulations, the importance of the distribution and orientation of fibres has been clearly stated, as well as the need of models and formulations that help structural designers to consider this type of predictive tools and numerical results. The fracture tests were reproduced and provided a remarkable degree of accuracy. The results of both the numerical simulations and experimental results have provided valuable discussion and verification of the parameters and expressions proposed in previous research [11, 42] as regards the influence of the coefficient of orientation.

This is a good paper!

Reviewer 3 Report

a) The paper addresses an interesting and relevant issue that is a pity that the authors have decided to focus on new plastic fibres instead of fibres obtained from recycled plastic. See for instance Pacheco-Torgal, F.; Khatib, J.; Colangelo, F.; Tuladhar, R.. eds. 2018. Use of recycled plastics in eco-efficient concrete ed. 1. Elsevier Science and Technology, Abington Hall, Cambridge

b) The authors need to supply a new fig 1 with a fibre image having a higher resolution

c) Does the different SP content was meant to insure the same workability level ?

d) Does the homogeneous distribution of fibres suggested by the authors has anything to do with the real conditions of fibre distribution ? Does this test can really assess the optimum reinforcement ?

e) Why a fibre dosage has been fixed at 6kg/m³ ?

f) Some text is confuse see for instance text in line 435 of conclusions “appear at the same strain states, they compete which”

g) Reference 1 is outdated. Cite instead the paper: Pakravan, H. R., Latifi, M., & Jamshidi, M. (2017). Hybrid short fiber reinforcement system in concrete: A review. Construction and Building Materials142, 280-294.

Author Response

a) The paper addresses an interesting and relevant issue that is a pity that the authors have decided to focus on new plastic fibres instead of fibres obtained from recycled plastic. See for instance Pacheco-Torgal, F.; Khatib, J.; Colangelo, F.; Tuladhar, R.. eds. 2018. Use of recycled plastics in eco-efficient concrete ed. 1. Elsevier Science and Technology, Abington Hall, Cambridge

The authors appreciate the suggestion performed by the reviewer and will consider to widen their research lines in order to include fibres obtained from recycled plastic.

b) The authors need to supply a new fig 1 with a fibre image having a higher resolution

 The authors have provided a new version of the Figure 1 with a better resolution. Such figure can be seen right below these lines. 

c) Does the different SP content was meant to insure the same workability level ?

The authors thank the comment of the reviewer because the explanation of this aspect improves the total quality of the contribution. The following sentence has been included in the manuscript: 

The calcium carbonate content of the limestone powder was higher than 98%, with less than 0.05% being retained in the 45 µm sieve. In the SCC and in the VC alike, an admixture named Sika Viscocrete 5720 (a polycarboxylate-based superplasticizer with a solid content of 36% and 1090 kg/m³ density) was employed.The changes in the superplasticizer contents helped to obtain the self-compactability in the freshstate behaviour of SCC6-60 and SCC6-430.

d) Does the homogeneous distribution of fibres suggested by the authors has anything to do with the real conditions of fibre distribution ? Does this test can really assess the optimum reinforcement ?

 The comment of the reviewer deals with an interesting matter that deserves a further explanation. In order to clarify such comment the following wording has been added to the manuscript. 

Consequently, as the volumetric fraction that corresponds to such addition is 0.66%, the amount of fibres in a 100x100m² section is 66. The disposition of the fibres in the middle section of the specimens can be seen in Figure 2.The homogeneous distribution of fibres in the ligament of the cross section differ from those found in real practice as was determined in [18-24, 37-38]. Such homogeneous distribution has been considered numb to the pouring method, the fresh-state behaviour of concrete and the wall effect and consequently the fibres have been distributed homogeneously. Although, it is true that a better fracture behaviour could be obtained if all the fibres were placed in the two lower thirds of the ligament it should not be forgotten that such distribution would be highly rare and quite far from everyday practice. 

e) Why a fibre dosage has been fixed at 6kg/m³ ?

The reviewer has pointed out to an aspect of the research conducted that should be clarified. In order to so, the following sentence has been added to the manuscript: 

In order to address such shortcomings in the field of PFRC, an experimental study has been performed with self-compacting concrete and conventional vibrated concrete with a fibre dosage of 6kg/m3of 60mm-long polyolefin fibres randomly distributed and named SCC6-60 and VCC6-60 respectively. Additionally, self-compacting concrete (SCC) with the same amount of fibres ideally positioned with a homogeneous distribution in the cross section of the concrete elements was manufactured and termed SCC6-430. Previous experience in assessing the fracture behaviour of PFRC with different amounts of polyolefin fibres have suggested that the sensitiveness of the distribution of fibres is lower for dosages of fibres equal or above 10kg/m³. Moreover, for dosages of polyolefin fibres of 4.5 or 3 kg/m³ there were occasions when there were only few fibres in the fracture surface. Such phenomenon hampers assessing the influence of the fibre orientation and distribution of fibres in the fracture behaviour of concrete. Accordingly, a medium dosage of polyolefin fibres (6kg/m³) was employed in this research. The previous information might be of interests for future research and everyday production alike

f) Some text is confuse see for instance text in line 435 of conclusions “appear at the same strain states, they compete which”

The sentence cited by the reviewer has been rewritten in order to improve the readability of the present manuscript.

Applications: 

F.S. Khalid, J.M. Irwan, M.H. Wan Ibrahim, N. Othman, S. Shahidan, “Performance of plastic wastes in fiber-reinforced concrete beams”, Construction and Building Materials183, pp. 451-464, 2018

S. Yin, R. Tuladhar, F. Shi, M. Combe, T. Collister, N. Sivakugan, “Use of macro plastic fibres in concrete: A review”, Construction and Building Materials, 93, pp. 180-188, 2015.

F. Pelisser, A.B.D.S. Neto, H.L. La Rovere, R.C.D. Pinto, “Effect of the addition of synthetic fibers to concrete thin slabs on plastic shrinkage cracking”, Construction and Building Materials24 (11), pp. 2171-2176, 2010

Mechanical charaterisation 

A. Al-Hadithi, N. Naji Hilal, “The possibility of enhancing some properties of self-compacting concrete by adding waste plastic fibers”, Journal of Building Engineering, 8, pp. 20-28,2016 

R.P. Borg, O. Baldacchino, L.Ferrara, “Early age performance and mechanical characteristics of recycled PET fibre reinforced concrete”,Construction and Building Materials, 108, pp. 29-472016

N. Pešić, S. Živanović, R. Garcia, P. Papastergiou, “Mechanical properties of concrete reinforced with recycled HDPE plastic fibres”, Construction and Building Materials, 115, pp. 362-3702016

P. Pujadas, A. Blanco, S. Cavalaro, A. de la Fuente, A. Aguado, “Fibre distribution in macro-plastic fibre reinforced concrete slab-panels”, Construction and Building Materials, 64, pp. 496-503, 2014

Given that some of them appear at the same strain states, it is difficult to separate the influence of each of them.

g) Reference 1 is outdated. Cite instead the paper: Pakravan, H. R., Latifi, M., & Jamshidi, M. (2017). Hybrid short fiber reinforcement system in concrete: A review. Construction and Building Materials142, 280-294.

The authors appreciate the suggestion of the reviewer as an updated list of references will increase the robustness of the contribution. Thus, reference number 1 has been substituted by the one proposed by the reviewer.